# A Superimposed QD-Based Optical Antenna for VLC: White LED Source

**DOI:** 10.3390/nano12152573

**Published:** 2022-07-27

**Authors:** Shaghayegh Chamani, Ali Rostami, Peyman Mirtaheri

**Affiliations:** 1Photonics and Nanocrystal Research Laboratory (PNRL), University of Tabriz, Tabriz 5166614761, Iran; shaghayegh_chamani98@ms.tabrizu.ac.ir; 2SP-EPT Laboratory, ASEPE Company, Industrial Park of Advanced Technologies, Tabriz 5169654916, Iran; 3Department of Mechanical, Electronics and Chemical Engineering, OsloMet—Oslo Metropolitan University, 0167 Oslo, Norway

**Keywords:** visible light communication (VLC), optical wireless communication (OWC), superimposed quantum dots, light-emitting diode (LED), Monte Carlo ray-tracing simulation, luminescent solar concentrator (LSC), optical receiver antenna, Li-Fi

## Abstract

Visible light communication (VLC) is a versatile enabling technology for following high-speed wireless communication because of its broad unlicensed spectrum. In this perspective, white light-emitting diodes (LED) provide both illumination and data transmission simultaneously. To accomplish a VLC system, receiver antennas play a crucial role in receiving light signals and guiding them toward a photodetector to be converted into electrical signals. This paper demonstrates an optical receiver antenna based on luminescent solar concentrator (LSC) technology to exceed the conservation of etendue and reach a high signal-to-noise ratio. This optical antenna is compatible with all colors of LEDs and achieves an optical efficiency of 3.75%, which is considerably higher than the similar reported antenna. This antenna is fast due to the small attached photodetector—small enough that it can be adapted for electronic devices—which does not need any tracking system. Moreover, numerical simulation is performed using a Monte Carlo ray-tracing model, and results are extracted in the spectral domain. Finally, the fate of each photon and the chromaticity diagram of the collected photons’ spectra are specified.

## 1. Introduction

The impressive growth of data traffic challenges our communication infrastructure [1]. Moreover, there is an essential need to dissolve the problem of spectrum scarcity and limited channel capacity in today’s exponentially growing wireless communication demand [2]. Many attractive characteristics of optical wireless communication (OWC) and visible light communication (VLC), such as real-time data transfer with high reliability, lower electromagnetic interference, more secure indoor wireless access, unlimited bandwidth, and better area spectral efficiency, make them promising successors of radio frequency (RF) communication [2,3]. VLC provides simple, flexible, and cost-effective OWC transmission [4] and is an auspicious technology for sixth generation (6G) networks because of the broad unlicensed spectrum [2]. Currently, there is a push toward green sustainable communication sources, such as VLC based on light-emitting diodes (LED) [5], which can support both lighting and communication simultaneously [4,6,7]. In other words, it modulates the intensity of light to transmit information at a very high rate [2,8]. The human eye cannot detect intensity modulation pulses and preserves them as a constant light [9]. Additionally, LEDs can be utilized in many fascinating applications, including localization, positioning [10,11], intelligent transportation systems [6], underwater communication [12], etc. LEDs have an extended lifetime, low power consumption (they conserve energy by 80% [13]), low cost, and a smaller carbon footprint in comparison with traditional lighting sources [14]. Currently, one of the world’s major concerns is the environmental impact of our choices. Due to serious environmental issues such as energy shortage, global warming, and greenhouse gas emission [15,16,17], it is of great importance to utilize LEDs instead of other classic light sources. Furthermore, the high switching frequency and high response sensitivity of LEDs make them a more attractive light source for VLC applications [4,6].

Based on the factors mentioned above, VLC based on LEDs has drawn remarkable scientific research attention. However, the chief focus of related research is usually on demodulation technology, LED arrangement, and channel coding but not on the optical receiving system [6,18,19]. Proper design of the optical receiving structure is an inevitable aspect of accomplishing the design of a VLC system [20], and a well-designed optical receiving antenna can ameliorate optical gain, optical receiving power, and signal-to-noise ratio (S/N) [6,21]. Optical receiving antennae are typically applied to achieve higher optical concentration in comparison to a bare photodetector [6,18]. As traditional common optical receivers, these models can be mentioned: first, the Fresnel lens, which limits the incidence angle of light [22,23]. Second, the Cassegrain antenna has a small field of view (FOV) and does not suit the VLC system [24]. Third, spherical and aspherical lenses (for example hemispherical lenses) have wide FOV but small gain [3,6]. Fourth, a compound parabolic concentrator (CPC) provides either small gain with wide FOV or significant gain with limited FOV [23,25]. Moreover, some limitations of geometrical optics restrict the performance of the optical antenna, including conservation of etendue. Conservation of etendue proclaims that there is a trade-off between maximum concentration gain of an optical system (*C*_max_) and FOV according to $C_{\max}=n^{2}/\sin^{2}(\theta )$ (where *n* is the refractive index of the optical system and θ is the acceptance angle, which describes FOV) if the wavelength of the optical system does not change [26,27,28]. Nevertheless, simultaneously obtaining both large FOV and high concentration gain is possible when the wavelength of the optical system changes [27]. In this regard, recently, a luminescent solar concentrator (LSC) was applied as a reassuring receiver technology for VLC [1]. The LSC principally emerged to collect solar energy efficiently and guide it toward side-attached solar panels. Moreover, thanks to its unique features, including energy down-conversion and light concentration, the LSC has become a versatile photonic technology platform garnering significant interest for applications such as VLC, compact dark-field imaging, and multistate smart windows (14.0). The LSC is a waveguide device for photon management with active optical centers (fluorophores) which can absorb a specific range of wavelengths and emit them with longer wavelengths (depending on its absorption and emission spectra designed). This down-conversion is known as the Stokes shift [29].

The LSC concept arose for the first time about 40 years ago to overcome the problems of photovoltaic cells in areas such as cost, efficiency, simplicity, aesthetics, etc. There is a highly significant interest in this aspect of the LSC. Extensive work has been carried out. However, attractive applications of LSC technology are not exclusive to this aspect. They can currently be employed as greenhouse coatings [30], microreactors [31], image recording and of movement detection technology [32], for dark-field imaging [33], and as VLC systems [27]. Only a few works have been done over the last decades in the context of VLC. In 2016, Manousiadis et al. demonstrated a planar LSC antenna doped with single-type dye and sandwiched between two microscope slides. The resulting data rate and the FOV were reported as 190 Mb/s and 60°, respectively, in this work [26]. Simultaneously, Peyronel et al. designed a ball-shaped fiber-based LSC antenna, and the data rate and FOV were reported to be 2.1 Gb/s and 3.9π, respectively [34]. Recently, in 2022, Chamani et al. proposed a quantum-dot-doped receiver antenna based on LSC technology with novel semi-sharp geometry through which an optical efficiency of 1.058% was reported. With this backdrop, a lot of room for improvement can be seen in this area. On a practical scale, LEDs as illuminators are not necessarily a specific color (e.g., blue), but they are usually a combination of various wavelengths within the visible range. Hence, matching the receiver antenna with the entire visible spectrum (i.e., white LED) can play a key role in VLC mutation. Additionally, the higher the efficiency, the better the performance, so it is of great importance to improve the efficiency of the antenna.

In addition, theoretical simulation is a momentous method for analyzing performance and confirming the parameters of the LSC prototype [29]. Since its adoption, a wide range of models, including theoretical, thermodynamic, and light tracing models, have been developed to scrutinize the behavior of the LSC and its parameters [35]. Among all methods, the Monte Carlo ray-tracing model as a numerical method affords diverse possible outcomes, greater flexibility, and a presumptive nature and can be extensively used in most QD-based LSC devices [36]. It has a cogent usage in situations wherein no deterministic algorithm exists and variables have coupled degrees of freedom. It is the most generally popular in other sciences, such as statistics, physics, finance, engineering, mathematics, and project management [37].

In this work, we propose an optical antenna based on an LSC with three active optical centers (generally superposition of quantum dots (QDs)) with different materials and sizes to absorb photons from white LEDs and then guide re-emitted photons to the edge of the LSC to be captured by the photodetector. This work is inspired by Ref. [1], which uses the novel proposed semi-sharp schematic of the antenna. Ref. [1] has demonstrated that first, the presented antenna is adaptable to electronic devices and integrating systems due to its tiny dimensions and relatively high efficiency. Second, it allows for the attachment of a very small photodetector on the tip of the antenna due to its shape (a small photodetector is sensitive, fast, and cost-effective). Here, not only are all these advantages present, but we also emphasize matching the antenna with white LEDs and increasing optical efficiency. Since white LEDs have such a broad spectrum that only a fraction of their spectrum is absorbed by one type of QD, we have used three various QDs. One of the most prevailing loss mechanisms in the LSC is mismatching between light spectrum and active optical centers because the mismatched part of the spectrum remains unabsorbed and consequently does not become involved in output efficiency [38]. Additionally, efficiency is the most dominant factor in measuring the performance of this system [38]. To address these problems, we apply the superposition method to cover the white LED spectrum and increase efficiency as well. We analyze the proposed antenna employing the Monte Carlo ray-tracing model to evaluate the optical efficiency of this LSC device based on three types of QDs and calculate the fate of each photon, the percentage of every event, and eventually extract corresponding spectra. Furthermore, we compute the chromaticity coordinates of the light spectrum which reaches the photodetector using the CIE 1931 standard observer.

The rest of the paper is organized as follows. Section 2 describes the operating method, structure, and components of the antenna and presents doped QDs and physical processes in the antenna. Section 3 delineates the simulation method of the antenna and its corresponding steps. Then, the obtained results are discussed in Section 4. Finally, Section 5 provides a summary.

## 2. Definition and Operating Principles of the LSC

LSC technology in the field of communication works on the principle of collecting modulated light incident on a large area and then converting it to longer wavelengths and directing the newly generated radiation to an optical detector with a small area [39]. It is generally comprised of a transparent matrix doped with emissive materials (fluorophores). Incident light can be absorbed by fluorophores and re-emitted at distinguished wavelengths matching the photodetector’s operating spectral region. The re-emitted light density is trapped inside and directed to the edges of the waveguide through total internal reflection (TIR) and harvested by a photodetector coupled to the one small edge of the matrix. Finally, the modulated light signal is converted to an electrical signal which carries data [39,40,41,42,43,44]. Since the surface area of the LSC exposed to light signal is substantially higher than the edge area, it can concentrate light and improve the flux radiation incident onto the photodetector edge [45,46]. Additionally, the LSC does not require any tracking system because it exploits efficiently diffused radiation as well as direct radiation [1].

Up to now, different approaches have been considered to enhance the performance of the LSC [29]. Emissive material used for the light conversion is the most indispensable entity to be considered in this regard [41]; additionally, its emission peak should overlap with the maximum of the spectral response of the photodetector [47]. Vastly different types of fluorophores can be classified into organic dyes [48], lanthanide complex hybrids [49], perovskites [50], and quantum dots (QDs) [37]. Approximately four decades ago, organic dyes such as Lumogen Red, Rhodamine B, Coumarin, etc., were a natural and high-performance choice for LSCs [41]. However, despite having upsides—such as high photoluminescence quantum yield (PLQY) (i.e., the number of emitted photons per absorbed photons in the luminescent material [1]), large absorption coefficient, easy availability, good solubility, and low cost [41]—organic dyes suffer from narrow absorption spectra and unfriendly photodegradation [1]. During the last decade, QDs have been pursued as an outstanding fluorophore for LSCs thanks to their excellent spectral tunability, highly efficient photoluminescence, stability, broadband absorbance, and high quantum yield (QY), which is defined by the ratio of the emitted photons to the absorbed photons in the QD [38,51].

Here, we analyze an LSC as an optical receiver antenna which is proposed to have a semi-sharp schematic and small dimensions, as shown in Figure 1a,b. The host matrix is made up of polyvinyl alcohol (PVA), and three types of core–shell QDs are doped in it as fluorophores (see Figure 1c). Superpositioned core–shell QDs are Se/SiO_2_, CdS/SiO_2,_ and Au/TiO_2,_ and their absorption and photoluminescence spectra—which are obtained experimentally based on the QDs’ size and synthesis method to be used as input parameters in the Monte Carlo simulation method—are shown in Figure 2a–c. Furthermore, white LED is utilized as the light source of this system and is irradiated onto the front surface of the antenna. Its spectrum is shown in Figure 2d, which is available in a commercial white LED datasheet. Each QD is responsible for absorbing a fraction of the visible light region according to its size and synthesis method; thereby, the totality of their effects results in absorbing the whole spectrum of the white LED. Generally, the incoming photon is absorbed by one of the types of QDs depending on the photon’s wavelength, which is within the related QD absorption range. Then, each type of QD re-emits the photon at the corresponding photoluminescence wavelength. Next, the TIR phenomena cause the re-emitted photon to be trapped inside the waveguide and ultimately guided toward the edges of the structure. The photon that is harvested by the tip-mounted photodetector (see Figure 1b) is converted to an electrical signal for data transmission purposes.

Different phenomena may take place in the LSC structure, as illustrated in Figure 3, when a light beam and waveguide interact [1,38]. Firstly, due to the difference between the refractive index of the surrounding medium and waveguide, a photon may be reflected from the top surface even without entering the waveguide (a). After a photon enters the LSC, there are two possibilities: the photon passes through the waveguide without being absorbed (b) or it is absorbed by one type of QD (c). Rarely, there is a small probability that the host matrix absorbs and quenches the photon; nevertheless, this is ignored in this paper (d). Then, a QD that absorbed the photon may emit it (e) or not, depending on the QD’s PLQY. Quenched photons are known as non-radiative recombination (f). On the other hand, emission angle and direction, which are determinative in the fate of the re-emitted photon, are also calculated randomly. If the angle is greater than the critical angle, the photon remains in the waveguide due to TIR (g); otherwise, the photon exits the waveguide, which is known as escape-cone loss (h). Furthermore, after some TIR, the photon may reach the photodetector, or there is a small probability that the re-emitted photon directly reaches the photodetector without any prior TIR. Additionally, the re-emitted photon, depending on its new wavelength, may be reabsorbed by either the same QD or the other 2 types of QDs (i); this mechanism is known as reabsorption loss. The process can be analyzed by the Monte Carlo ray-tracing simulation, depicted in the next section.

## 3. LSC Monte Carlo Ray-Tracing Simulation

Monte Carlo ray-tracing simulation specifies the ultimate fate of all the incoming photons based on events such as reflection, transmission, absorption It also specifies emission processes based on mathematical equations such as Fresnel’s law, Snell’s law, and the Beer–Lambert law and empirical data such as absorption spectrum, photoluminescence spectrum, and PLQY of fluorophores [1,38,52]. The flow chart used to develop the Monte Carlo simulation is presented in Figure 4. Each time the algorithm is run, only one randomly generated photon is considered. The algorithm aims to detect whether a single photon will be lost or harvested to evaluate the optical efficiency of the device. Additionally, types of losses—including reflection, transmission, non-radiative recombination, and escape-cone loss—can be determined. In the beginning, 100,000 photons take part in the simulation and are assumed to distribute uniformly on the top surface of the antenna. The wavelength of each photon is sampled from the white LED spectrum (see Figure 2d). This sampling procedure starts with generating a probability density function (PDF) [38] of the white LED, which is directly generated from the emission spectrum of the white LED (the only difference is its normalization to the area of the curve, as written in Equation (1)). The PDF of the white LED is shown in Figure 5a. Next, the PDF is converted to the cumulative distribution function (CDF) [38], which is illustrated in Figure 5b, through which wavelengths of photons are sampled using the inverse transform sampling method [1].
PDF = (Distribution function/Area of the curve)(1)

The sampled photons are assumed to have uniform distribution and normal incident on the top surface of the LSC. On the contrary, due to the antenna’s schematic, the top inclined surface will be exposed to incident light with a non-zero input angle, as shown in Figure 3. This angle is calculated by trigonometric relations to be used during the simulation. Additionally, during the simulation, the white LED is assumed to be TE-polarized. To determine the reflection loss, the reflection equation in Fresnel law is used [53]. Perspicuously, the difference in the refractive indices of two mediums causes reflection loss, which is calculated by the Fresnel reflection equation for TE polarization [53], as per Equation (2):(2)$$R=R_{s}={\left(\frac{n_{Air}\cos(\theta_{i})-n_{LSC}\cos(\theta_{t})}{n_{Air}\cos(\theta_{i})+n_{LSC}\cos(\theta_{t})}\right)}^{2}={\left[\frac{n_{Air}\cos(\theta_{i})-n_{LSC}\sqrt{1-{\left(\frac{n_{Air}}{n_{LSC}}\sin(\theta_{i})\right)}^{2}}}{n_{Air}\cos(\theta_{i})+n_{LSC}\sqrt{1-{\left(\frac{n_{Air}}{n_{LSC}}\sin(\theta_{i})\right)}^{2}}}\right]}^{2}$$
where *n_Air_* is the refractive index of air and equals one, *n_LSC_* is the refractive index of PVA. It is assumed to be constant during the simulation (*n_LSC_* ≈ 1.49 for the visible region) [54]; *θ_i_* and *θ_t_* are incidence and transmission angles, respectively. For photons irradiated to the flat top surface of the antenna which have zero incidence angle (see Figure 3), Equation (2) is simplified to Equation (3). Thus, approximately 4% of initial photons are removed from the rest of the process as reflection loss.
(3)$$R={\left(\frac{n_{Air}-n_{LSC}}{n_{Air}+n_{LSC}}\right)}^{2}$$

Then, it should be clear whether the QDs absorb the photon or not. For a single photon, initially, it should be investigated which type of QD (type A, B, or C) may absorb the photon. A QD must have these conditions to have a chance to absorb the photon: firstly, the photon’s wavelength must be valid within the absorption range of the QD. Secondly, in that specific wavelength, the QD must have the largest absorption coefficient (based on the absorption spectrum which is shown in Figure 2a–c). Next, the Beer–Lambert law determines the probability of a photon being absorbed after moving along a distance, Δ*d* (in cm) [52]:(4)$$A_{t}=1-10^{-\varepsilon_{t}(\lambda )C_{t}\Delta d},\;t=QD_{A},\;QD_{B},\;QD_{C},\;$$
where *t* is the type of the QD; *ε_t_*(*λ*) is the extinction coefficient of QDs (i.e., absorption spectrum of a QD which is photophysical characterization); and *c_t_* is the concentration of QDs, which must match the unit of *ε_t_*(*λ*). According to the QD that absorbed the photon, Δ*d* is calculated by Equation (4), which is specific to the same QD, and the corresponding *ε_t_*(*λ*) and *c_t_* are applied.

Moreover,
(5)$$\Delta d=-\frac{\log(1-A_{t})}{\varepsilon_{t}(\lambda )C_{t}}$$

In the simulation, absorption probability is assessed by a randomly generated number *ζ* within [0, 1], as per Equation (6).
(6)$$\Delta d=-\frac{\log(\xi )}{\varepsilon_{t}(\lambda )C_{t}}$$

The photon will be absorbed by calculating the *A_t_* coefficient, and if Δ*d* is found to be smaller than the thickness of the LSC, its (x,y,z) position is stored for the rest of the process. Otherwise, it will cross through the LSC directly, which is referred to as transmission loss. In the next step, it should be determined whether the QD would emit the absorbed photon or not. This is performed based on the QY, which is defined by the ratio of emitted photons to absorbed photons [1]. It is compared to a randomly generated number (*β*) which is within [0, 1]. For values of *β* lower than the QY, the photon will be re-emitted, and its wavelength is selected from the CDF of the emission spectrum of the QD which had absorbed the photon (type A, B, or C). Figure 6 shows the PDF (obtained directly from emission spectra of QDs using Equation (1)) and the CDF for the emission spectra of QDs adapted from their PDF. Additionally, the emission angle is randomly chosen (uniform distribution) to update the (x,y,z) position of the newly emitted photon. The distance that the re-emitted photon crosses before being reabsorbed again is given by Equation (7) [52].
(7)$$\Delta d=-\frac{\log(\beta )}{\varepsilon_{t}(\lambda )c_{t}}$$
where *β* is the random number which is generated in this step to determine the emission probability of the absorbed photon.

Next, it should be checked if the photon is inside the LSC or interacts with surfaces. If yes, it has a chance to be reabsorbed, and if not, it should be clear which surface the proton interacts with. The photon will be harvested if it impacts the edge on which a photodetector is attached. Otherwise, two scenarios may take place: first, the photon is reflected because of TIR, and it may be reabsorbed or interact with other surfaces. Second, the photon runs away from the surface, and inconsequence loss, which is known as escape-cone loss, occurs. This process is repeated for all 100,000 initial photons to trace their fates. To evaluate the performance of the LSC, optical efficiency (OE) is defined in Equation (8) [52].
η_opt_ = (Collected Photons/Incident Photons)(8)

Furthermore, geometric gain, another parameter of the LSC which measures the maximum possible photon flux concentration when all the other factors are perfect, is determined as per Equation (9) [52].
(9)$$G=\frac{A_{LSC}}{A_{PD}}$$
where *A_LSC_* is the total area of the top surface of the LSC (both flat and inclined) and *A_PD_* is the area of the located photodetector.

## 4. Results and Discussion

To realize white LED spectrum absorption coverage with a receiver optical antenna, core–shell QDs of Se/SiO_2_, CdS/SiO_2_, and Au/TiO_2_ materials have been applied based on LSC technology. We have developed sizes and core radiuses of QDs experimentally to design their photophysical characterization, such as absorption and photoluminescence spectra, as shown in Figure 2a–c. As illustrated, each type of quantum dot can absorb a wide fraction of the commercial white LED spectrum (see Figure 2d) and convert it to lower energy via Stokes shift. As a result, the entire range of the visible spectrum has been covered completely. This gives our antenna a greater degree of freedom to work with any color of LED used in ambient lighting. Additionally, Se/SiO_2_, CdS/SiO_2_, and Au/TiO_2_ emit the absorbed photons at around 400 nm (violet), 520 nm (blue-green), and 600 nm (orange), respectively. In Monte Carlo ray-tracing simulation, many parameters are essential as input parameters, including LSC waveguide geometry and dimensions, absorption spectrum, emission spectrum, concentration and QY of each type of QDs, and refractive indices of surrounding medium and waveguide. QY differs between 0 and 1 based on the QDs synthesis method, and here we found 0.9 for all three types. Additionally, we considered the concentration of QDs to be approximately 120 PPM. Since the concentration of QDs is not enough to change the refractive index of PVA, we have assumed the refractive index of the waveguide is equal to that of PVA and constant throughout the study (*n_LSC_* = 1.49 for the visible region). Moreover, the dimensions of the designed antenna are considered as shown in Figure 1. By accomplishment of the simulation, based on Equation (8), the optical efficiency of the antenna is obtained at 3.75%, which is significantly higher than the optical efficiency of the similar LSC antenna in Ref [1] with single-graphene QD and other planar examples. This means that we have reached optical efficiency enhancement compared to the LSC antenna with similar schematics. At the same time, geometric gain plays a key role in the overall performance of the antenna, and its aspect ratio is computed to be 44 using Equation (9). Based on simulation results and the fate of 100,000 initial photons determined by Monte Carlo ray-tracing simulation, 4.850% of initial photons are reflected from the top surface of the LSC; this is called reflection loss. After entering, 1.546% of photons pass through directly, which is known as transmission loss. The remaining 93.604% of photons are absorbed by types of QD depending on their wavelengths and absorption coefficients. Another 16.096% of absorbed photons are not emitted due to non-radiative recombination. Simultaneously, while 3.755% of photons reach the photodetector, 36.110% exit from other remaining edges of the LSC. Meanwhile, we have extracted the spectra of all the events in Figure 7. Here, we used Monte Carlo ray-tracing simulation to predict the features and performance of the antenna, which is able to track photons and determine the fate of all the initial photons and eventually calculate the likelihood of each event that occurred. Thus, after completing the simulation for all 100,000 initial photons, it is clear how many photons with which wavelengths have reflected/transmitted/absorbed/collected, etc. Thereby, in Figure 7, the number of photons and their corresponding wavelengths in each event inside the antenna are illustrated. At first, the generated spectrum of the initial random photons from the irradiance spectrum of a commercial white LED (Figure 2d) is exhibited in Figure 7a. Then, spectra of the reflection loss and transmission loss are presented in Figure 7b,c, respectively. Following this, the spectrum of the absorbed photons is shown in Figure 7d. Then, regarding the ultimate fate of the absorbed photons, the spectrum of the quenched photons by non-radiative recombination is presented in Figure 7e, and spectra of the beneficial collected photons and photons which exited from other edges are illustrated in Figure 7f,g, respectively. The existing shoulders in Figure 7f,g are because of reabsorption loss which occurs in the LSC. We summarize all the spectra in Figure 7h.

Our approach in this study and the conclusive intention of designing a receiver antenna using LSC technology involves several factors: first, using LEDs as a light source provides low energy consumption, less cost, higher lifetime, high data rate, and smaller carbon footprint. Second, using LSC definition for the main structure of the antenna to achieve high concentration gain (because of Stokes shift) and wide FOV without any need for a tracking system. Third, applying the schematic proposed in Ref [1], to create a small, sensitive, fast, and cost-effective antenna. It is worth noticing that our structured antenna has a small size, allowing it to be embedded inside electronic devices such as mobile phones, tablets, smart watches, quadcopters, virtual reality headsets, and even clothing, which enables rapid mobile communication and integration systems. It is sensitive, fast, and cost-effective because a small photodetector is located in the tiny end facet of the waveguide thanks to this schematic. Fourth, using three types of core–shell QDs to cover the visible spectrum allows the antenna to work with all colors of LEDs and enhances the optical efficiency (OE) of the antenna. The receiver antenna is simulated using the Monte Carlo ray-tracing model, the likelihood of each event that occurred is calculated, and corresponding spectra are extracted. On the last attempt, we computed the chromaticity diagram [55] of the white LEDs’ emission spectrum (see Figure 2d), as shown in Figure 8a, and the chromaticity diagram of harvested photons’ spectrum (see Figure 7f), as represented in Figure 8b. Figure 8 is a visual image to show the color of the input light and the output light of the antenna and their deference. As it is clear, there is a very minor difference between them, so we have regenerated the white LED’s color using the proposed antenna. Thus, a white light photodetector can be mounted on the tip of the antenna. Consequently, both the antenna input light spectrum and the antenna output light spectrum possess more degrees of freedom, and they are not limited to a narrow spectrum.

## 5. Conclusions

In conclusion, we have indicated an optical receiver antenna with LSC technology to create a small, fast, sensitive, cost-effective, and simple antenna for VLC proposes. Using LSC technology leads to overcoming the conservation of etendue. Hence, wide FOV, high concentration gain, and eventually high signal-to-noise ratio are accessible simultaneously. Furthermore, there is no need for any tracking system because the LSC collects both direct and diffused light. This antenna was configured with PVA as a host matrix and three forms of QDs in it to make it adaptable to the entire visible light spectrum and all colors of LEDs and ameliorate the efficiency of the antenna. It absorbs incident light, converts it to lower energy via Stokes shift, and guides it toward the photodetector. To model the antenna, the Monte Carlo ray-tracing method was developed, and results were reported in percentage and spectral domains. Moreover, since three types of QDs were applied, it was shown that this antenna could absorb the white LED spectrum, and an optical efficiency of 3.75% was obtained. It is worth mentioning that the proposed antenna has more degrees of freedom in terms of input light and output light, and it is not limited to a narrow specific spectrum. Furthermore, thanks to its small size, it can be incorporated into electronic devices easily.

## Figures and Tables

**Figure 1 nanomaterials-12-02573-f001:**
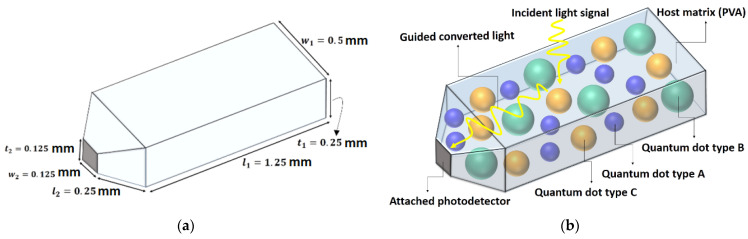
The proposed schematic of the considered optical antenna. (**a**) Dimensions of the considered antenna. (**b**) Configuration of components in the antenna (host matrix, doped quantum dots, attached photodetector, incident light, and guided light). (**c**) Types of the core–shell quantum dots which are doped inside the antenna’s structure.

**Figure 2 nanomaterials-12-02573-f002:**
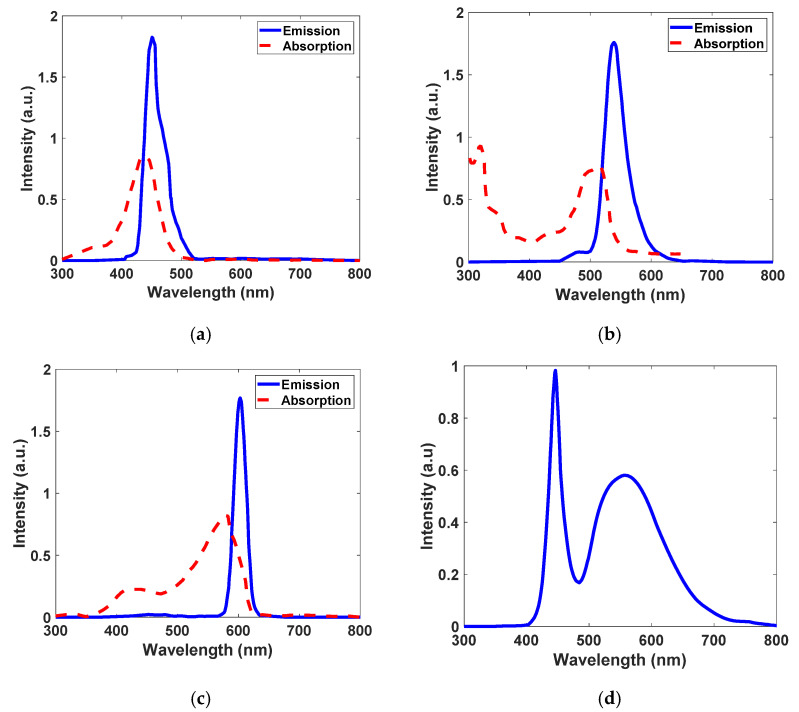
Experimentally obtained absorption and photoluminescence spectra for quantum dots and emission spectrum for commercially available white LED. (**a**) Absorption and emission spectra for quantum dot type A: Se/SiO_2_. (**b**) Absorption and emission spectra for quantum dot type B: CdS/SiO_2_. (**c**) Absorption and emission spectra for quantum dot type C: Au/TiO_2_. (**d**) Emission spectrum for commercial white LED.

**Figure 3 nanomaterials-12-02573-f003:**
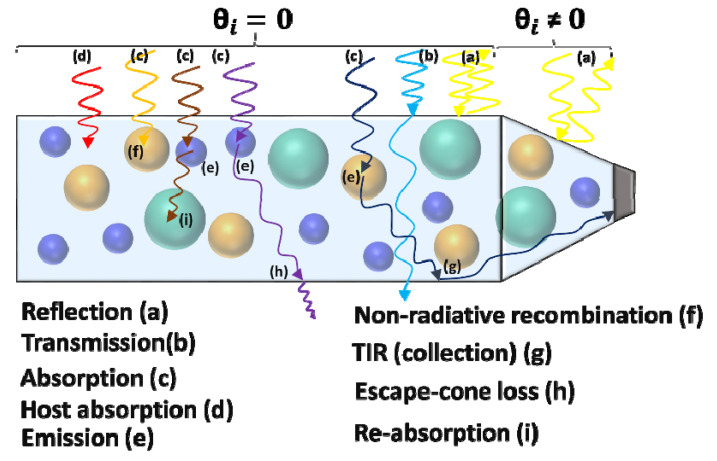
Side view of the antenna and different possible events that may take place inside the antenna when the light is irradiated to the upper surface of the antenna.

**Figure 4 nanomaterials-12-02573-f004:**
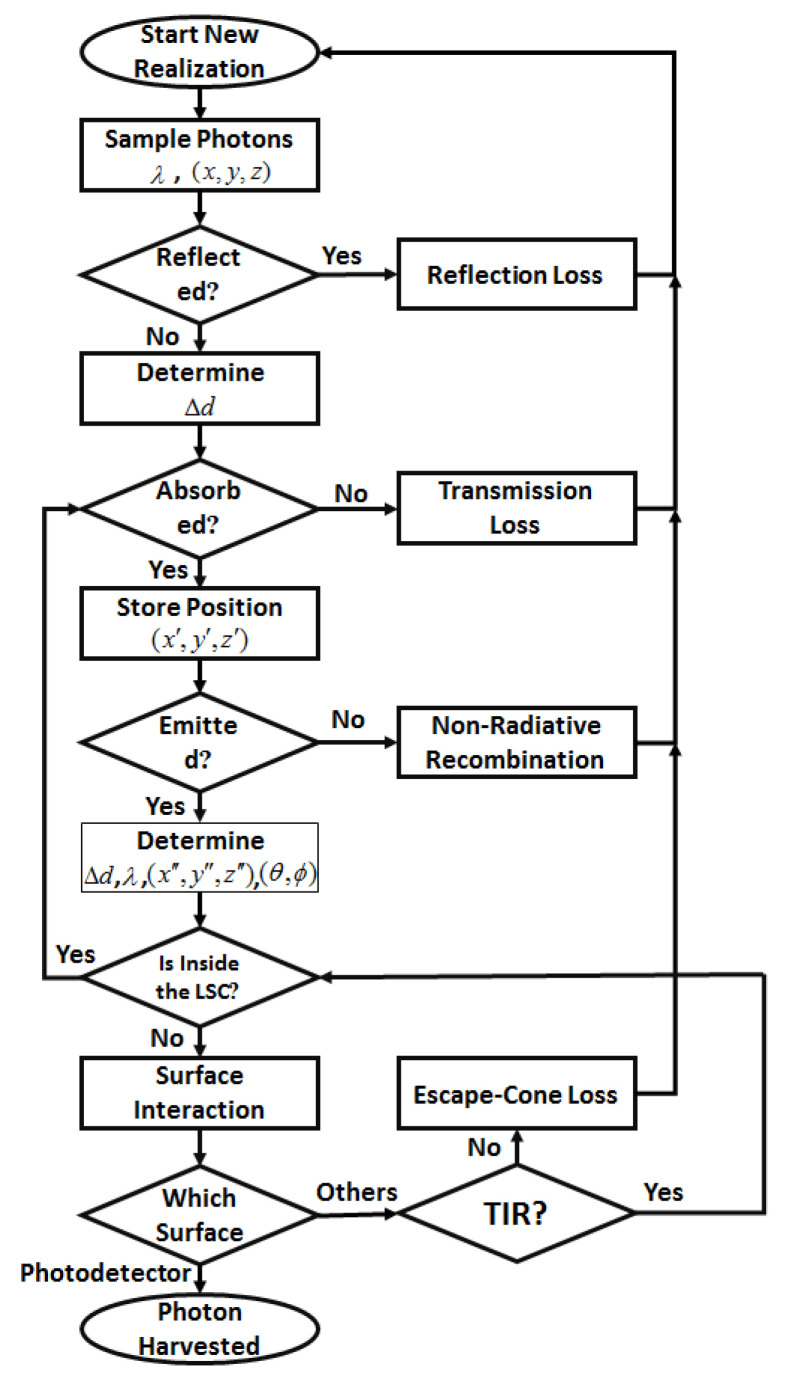
Monte Carlo ray-tracing algorithm that applied for numerical simulation.

**Figure 5 nanomaterials-12-02573-f005:**
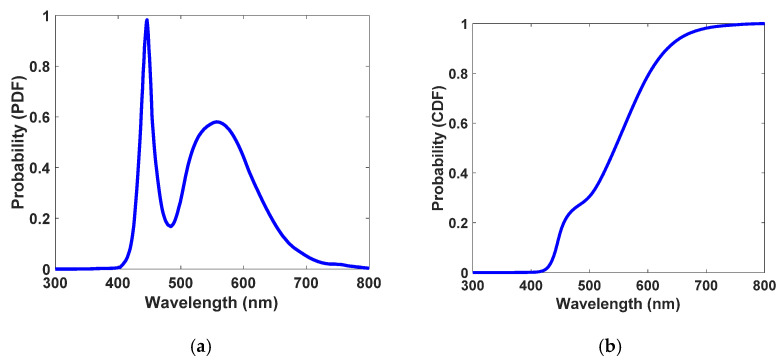
(**a**) Probability density function (PDF) for white LED. It is directly generated from the emission spectrum of white LED (the only difference is its normalization to the area of the curve). (**b**) Cumulative density function (CDF) for white LED adapted from its PDF curve.

**Figure 6 nanomaterials-12-02573-f006:**
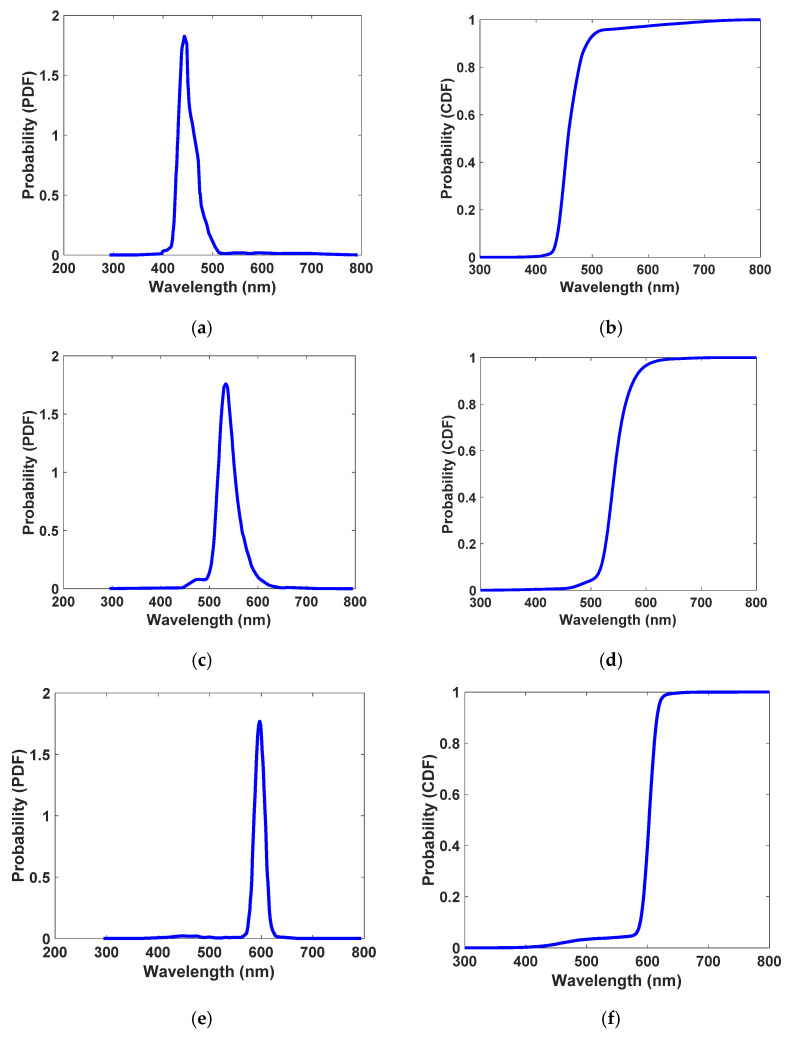
Probability density function (PDF) and cumulative density function (CDF) for quantum dot type A: Se/SiO_2_ (**a**,**b**), quantum dot type B: CdS/SiO_2_ (**c**,**d**), and quantum dot type C: Au/TiO_2_ (**e**,**f**), respectively.

**Figure 7 nanomaterials-12-02573-f007:**
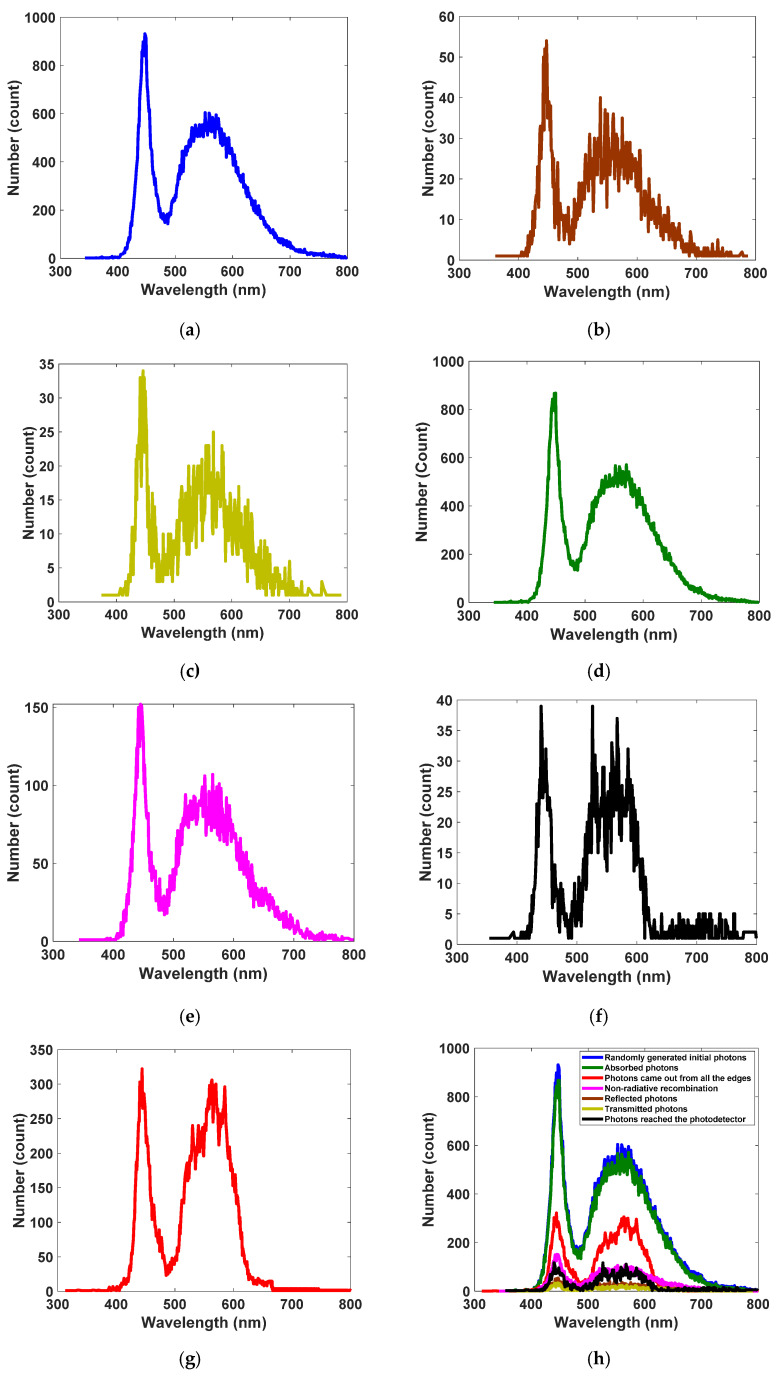
The obtained results in the form of spectra (number of the photons with their corresponding wavelengths) for each event occurred inside the antenna after completing the Monte Carlo simulation. (**a**) Sampled incident photons (100,000 randomly generated photons). (**b**) Photons reflected from the top surface. (**c**) Photons transmitted through the antenna without being absorbed. (**d**) Photons absorbed by three types of quantum dots. (**e**) Photons quenched by non-radiative recombination. (**f**) Photons that exited from all the edges of the antenna. (**g**) Photons collected by the photodetector. (**h**) All events in one figure.

**Figure 8 nanomaterials-12-02573-f008:**
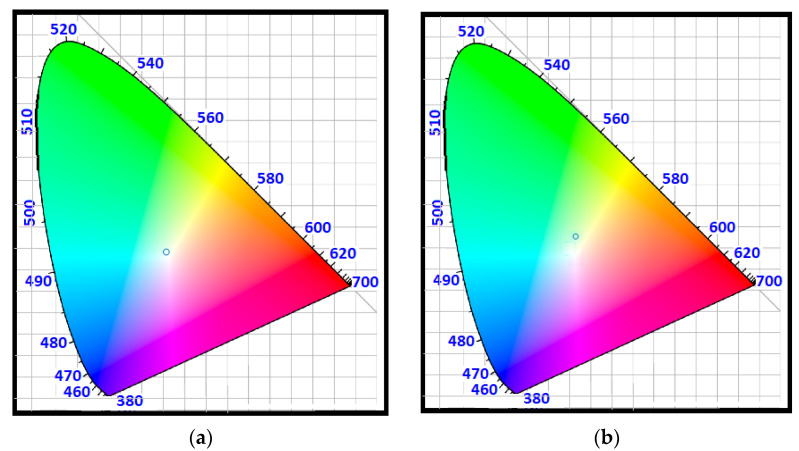
(**a**) White LED’s spectrum and (**b**) collected photons’ spectrum in the CIE 1931 standard observer chromaticity diagram.

## Data Availability

Not applicable.

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
