# Peer review of "A Superimposed QD-Based Optical Antenna for VLC: White LED Source"

_nanomaterials, 2022, doi:10.3390/nano12152573_

Round 1

Reviewer 1 Report

In this work, the authors have carried out a study of designing optical receiver antenna using the luminescent solar concentrator technique. The authors aimed to demonstrate the demonstrated devices with the features such as small scale, energy efficiency, low cost. According to their experiment and simulations, the authors claimed that the demonstrated antenna can absorb the white LED spectrum with efficiency of few percentages.

After carefully going through the manuscript, I have to say that a major revision must be made before a decisive decision can be reached. There are so many issues appearing in the manuscript. In the following, I will specify these one by one:

(a)  First of all, the presentation must be improved and the English must be polished. So far as I can see, there are many many grammar issues. I strongly suggest the authors to make a substantial revision on the presentation.

(b)  The authors have messed up most of the figures in the main text. These include the mislabeling of figures, the lack of explanations of the reported figures, the incomplete information on figure plots, and the incomplete figure plot. In particular, the absorption spectrum was missing from Figure 2(d) in page 5. Figure 4 in page 7 was not even cited in the manuscript. I don't understand how to read Figure 8 in page 13. Moreover, for Figures 2 (page 5), Figure 5 (page 7), Figure 6 (page 9), Figure 7 (pages 11 and 12), and Figure 8 (page 13), what are the parameters used in these plots? I am very curious to know how the authors measured photon numbers in Figure 7. More detailed explanations are necessary for the understanding of the data. Of course, other Figures mentioned above should also be further explained in the text, especially more information on their experiment including parameters should be provided. The discussions in page 10 must be revised as there are many misleading statements.

(c)  The described method in Session 3 is highly misleading and many errors can be observed here. First, the authors messed up the labeling of most of the equations quoted in this session. This mislabeling is also reflected from the very confusing descriptions in this part of the presentation. Among these equations, an explanation of Equation (1) in page 6 needs to be added. What is the equation quoted in page 7? As the first equation in page 8 is important to the reported work, more discussions are necessary. Especially, the involved epsilon_lambda and C_t are essential to the experiment. But the authors failed to provide the corresponding information on these parameters. In line 253, "At" should be "A_t". I do not see the essential difference between the third and forth equations quoted in page 8. The last equations quoted in this page are the same. I don't understand why the authors repeated them here. Most crucially, I don't understand the relation between the presented theory and the reported experiment. It looks like the authors simply presented an isolated theory and then reported separately an experiment. In the revision, I would like to see the connections between theory and experiment.

(d)  Some abbreviations are lack of explanations. For example, what is "CIE"? What does "PLQY" mean? What is "QY"? Also, the authors used "OE" to represent "optical efficiency" first and then used it to denote "device efficiency". Please correct these.

(e)  Many references are cited with incomplete information. For examples, Refs. [1-3, 5, 9, 13, 25, 45, 52] are incomplete. Please update them with the missing information. Please correct the author list in Ref. [23].

Author Response

Dear Editor

Thank you for allowing us to submit a revised draft of our manuscript entitled “The Superimposed QDs based Optical Antenna for VLC: White LED Source” to Nanomaterials. We appreciate the time and effort that you and the reviewers have dedicated to providing your valuable feedback on our manuscript. We are grateful to the reviewers for their insightful comments on our paper. We have been able to incorporate changes to reflect most of the suggestions provided by the reviewers. We have specified the changes within the manuscript using the track changes option in word. Here is a point-by-point response to the reviewers’ comments and concerns.

Bests

Ali Rostami

Reviewer 2 Report

In this work, Chamani et al. reported an optical receiver antenna with LSC technology to accomplish an efficient antenna for VLC systems. In the LSC technology, three types of core-shell QDs with different absorption and emission spectrum in the visible wavelength range were superimposed as the energy/wavelength down-conversion layer to improve the optical efficiency of the photodetector. Numerical simulation using Monte-Carlo ray-tracing model was performed to calculate the light transportation/absorption/reflection etc. processes. The results are valuable to guide future experiments for designing an efficient VLC system. The following questions need to be addressed before accepting.

1. The authors should check all figure numbers carefully. Line 184 should be Figure 2. Line 282 should be Figure 6.

2. Figure 1(d) is only the emission spectrum of the white LED.

3. The processes and legends in Figure 3 need to be reorganized. It’s better to follow the sequence of a->i from left to right. From lines 188-205, it’s better to mention the corresponding process during the description. For example, “a photon may be reflected from the top surface even without entering the waveguide (a)”.

4. The letter in a formula should be the same in the context. For example, in line 247, Ct should be ct. Please check related content carefully.

5. Line 312-314, the percentage of reflected and directly passed through photons is 4.85% and 4.54%, respectively. Why the percentage of remaining photons is 93.6%?

6. Are the results shown in Figure 2 experimental results? If so, did the author make such an antenna based on the proposed structure? Could you comment on the optical efficiency discrepancy in a real system and the simulated result?

Author Response

(The authors gave the same response as above.)

Round 2

Reviewer 1 Report

I have carefully gone through the revision as well as the response from the authors. As far as I can see, the authors do follow my suggestions and comments and make corresponding changes in the revised manuscript. The revised version is now more readable and accessible to a reader. The critical issues pointed out in the previous report have been addressed in the revision. Although the English can be further improved, at this stage I am happy to recommend it for publication in Nanomaterials.